Manuscript prepared for J. Name
with version 5.0 of the LaTeX class copernicus.cls.
Date: 20 May 2016

# A DNS study of aerosol and small-scale cloud turbulence interaction

**N. Babkovskaia[1], U. Rannik[1], V. Phillips[2], H. Siebert[3], B. Wehner[3], and M. Boy[1]**

[1]University of Helsinki, Department of Physics, Helsinki, Finland
[2]Lund University, Department of Physical Geography and Ecosystems Science, Sweden
[3]Leibniz Institute for Tropospheric Research, Leipzig, Germany

*Correspondence to:* N. Babkovskaia
(NBabkovskaia@gmail.com)

**Abstract.** The purpose of this study is to investigate the interaction between small-scale turbulence and aerosol and cloud microphysical properties using Direct Numerical Simulations (DNS). We consider the domain located at the height of about 2000 m from the sea level, experiencing transient high supersaturation due to atmospheric fluctuations of temperature and humidity. To study the effect of total number of particles ($N_{tot}$) on air temperature, activation and supersaturation we vary $N_{tot}$. To investigate the effect of aerosol dynamics on small-scale turbulence and vertical air motion we vary the intensity of turbulent fluctuations and the buoyant force. We find that even small amount of aerosol particles (55.5 cm$^{-3}$) and, therefore, small droplet number concentration strongly affects the air temperature due to release of latent heat. The system comes to an equilibrium faster and the relative number of activated particles appears to be smaller for larger $N_{tot}$. We conclude that aerosol particles strongly affect the air motion. In a case of updraft coursed by buoyant force the presence of aerosol particles results in acceleration of air motion in vertical direction and increase of turbulent fluctuations.

## 1 Introduction

Interaction of atmospheric turbulence with aerosol and cloud formation processes has been studied extensively. Due to non-linearity of particle formation and other aerosol dynamical processes, the fluctuations of temperature and relative humidity can have strong effect on aerosol formation. Large scale fluctuations of atmospheric properties, which occur for example in the atmospheric boundary layer, can be the drivers for initiation of particle formation (Easter et al., 1994). Mixing of air with different properties, including temperature and relative humidity, have been shown to enhance atmospheric nucleation significantly (Nilsson and Kulmala, 1998). More specifically, the atmospheric waves can increase the nucleation rate several orders of magnitude and affect also the size spectrum of the particles (Nilsson et al., 2000). Also the activation of atmospheric particles in cloud areas is affected by the fluctuation of supersaturation. Some droplets were shown to grow also in undersaturated conditions due to fluctuations and the bi-modal particle size distribution could be observed after initially unimodal particle population experienced fluctuating supersaturation (Kulmala et al., 1997).

In-cloud turbulence has been shown to intensify cloud-microphysical processes determining cloud properties (Benmoshe and Khain, 2014). Similarly, aerosol loadings also influence the numbers and sizes of cloud-particles, thereby influencing precipitation, cloud extent and hence the climate (Forster, 2007). Both turbulence and aerosol particles can influence the chemical reactions in the atmosphere providing surfaces for aqueous phase chemistry and promoting uptake of gaseous species. This can have repercussions for chemical reactions in the air. Yet both environmental factors, namely aerosol composition and turbulence, have been usually considered as separate influences.

The large-scale atmospheric turbulence is well known to affect aerosol processes significantly. In turn, aerosol properties may influence the buoyancy and turbulence. A motivation for a potential aerosol-turbulence interaction is that the most advanced cloud-microphysical models can now resolve the cloud edges where mixing occurs and where spatial gradients are great in turbulence and concentrations of hydrometeors, including aerosols. Consequently, by including a more complete set of such interactions in the models, the simulation of aerosols , turbulence and microphysics may be improved.

The main goal of this direct numerical simulation (DNS) study is to investigate interaction between small-scale tur-

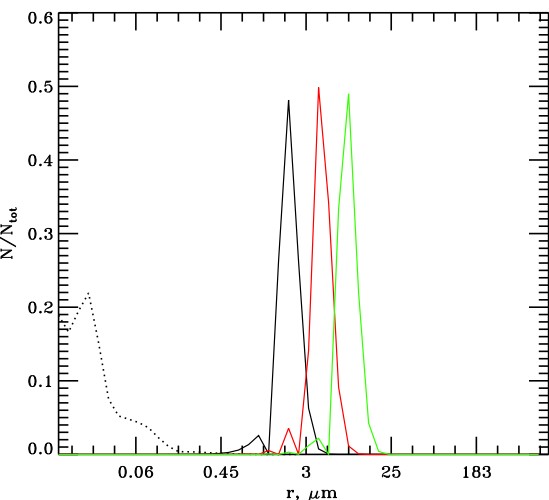

**Fig. 1.** Relative distribution of particles averaged over the domain at t=0 s (*black dotted curve*) and at $t = 3$ s (*solid curve* ) for the cases of $N_{tot} = 55.5$ cm$^{-3}$ (*green solid curve*), $N_{tot} = 555$ cm$^{-3}$ (*red solid curve*), $N_{tot} = 5550$ cm$^{-3}$ (*black solid curve*). Dotted curve corresponds to the observed distribution of aerosol; solid and dashed curves are distributions of droplets.

| | | |
|---|---|---|
| *case 1* | $N_{tot} = 55.5$ cm$^{-3}$ | $S_{av} = 10.3$ % |
| *case 2* | $N_{tot} = 555$ cm$^{-3}$ | $S_{av} = 10.3$ % |
| *case 3* | $N_{tot} = 5550$ cm$^{-3}$ | $S_{av} = 10.3$ % |
| *case 4* | $N_{tot} = 5550$ cm$^{-3}$ | $S_{av} = 0.6$ % |
| *case 5* | no particles | |

**Table 1.** The summary of key parameters for studying the effect of total number of particles on air temperature, supersaturation and activation. $S_{av}$ is initial supersaturation averaged over the domain.

| | equilibrium | non-equilibrium |
|---|---|---|
| *low intensive turbulence* | $T_0 = 285.4$ K  $f_0 = 10$ | $T_0 = 283.5$ K  $f_0 = 10$ |
| *high intensive turbulence* | $T_0 = 285.4$ K  $f_0 = 100$ | $T_0 = 283.5$ K  $f_0 = 100$ |

**Table 2.** The summary of key parameters for studying the effect of aerosol on turbulence.

bulence and aerosol and cloud microphysical properties. The chosen DNS domain is realistic for a small volume at cloud edge where turbulent mixing is a dominate feature (Katzwinkel et al., 2014) and can create fluctuations in temperature as well as humidity. Such conditions however appear transient because presence of atmospheric aerosols leads to depletion of supersaturation via condensation and droplet activation.

We use the high order public domain finite difference PENCIL Code for compressible hydrodynamic flows. The code is highly modular and comes with a large selection of physics modules. It is widely documented in the literature (Dobler et al., 2006; Pencil Code, 2001, and references therein). The chemistry module is responsible for the detailed description of the necessary quantities in a case of complicated chemical composition, such as diffusion coefficients, thermal conductivity, reaction rates etc. (Babkovskaia et al., 2011). The detailed description of aerosol module can be found in Babkovskaia et al. (2015). The paper is constructed as follows. Section 2 is devoted to the description of the model. Results are presented in Section 3. Section 4 provides the summary of our study.

## 2  Description of the model

*Aerosol*

As a starting point for our simulations we consider values typical for observations made in trade wind cumuli. During the CARRIBA project (Siebert et al. 2013, hereafter called SI13) total aerosol number concentrations in the marine sub-cloud layer up to a few 100 cm$^{-3}$ have been observed (Fig 6f in SI13). Although it was argued that the highest values are due to local biomass burning we consider a concentration of 550 cm$^{-3}$ as typical and 55 and 5550 cm$^{-3}$ as two extreme values for sensitivity test. The initial normalized aerosol number size distribution as shown as a dotted line in Fig. 1 compares well with the shape of the observed distribution shown as red line in Fig. 8 of SI13.

We assume a soluble aerosol (NaCl) which will dilute inside droplet. We take 50 size bins logarithmically distributed in the range [10 nm, 1000 $\mu$m]. As an initial distribution of particles we take the observational data at the sea level and assume that it is the same everywhere in the domain (see the distribution in Fig. 1). To analyze the effect of total number of aerosol particles, $N_{tot}$, on the structure and properties of turbulent motion we consider the following cases: $N_{tot} = 55.5$ cm$^{-3}$ and initial supersaturation averaged over domain $S_{av} = 10.3$ % (*case 1*); $N_{tot} = 555$ cm$^{-3}$ and $S_{av} = 10.3$ % (observed data, *case 2*); $N_{tot} = 5550$ cm$^{-3}$ and $S_{av} = 10.3$ % (*case 3*); $N_{tot} = 5550$ cm$^{-3}$ and $S_{av} = 0.6$ % (*case 4*); and no particles (*case 5*), see Table 1.

*Air composition*

We assume the following air composition O$_2$ + H$_2$O + N$_2$, where nitrogen mass fraction is taken to be $Y_{N_2} = 70\%$. The observations provide us the absolute humidity to set the ini-

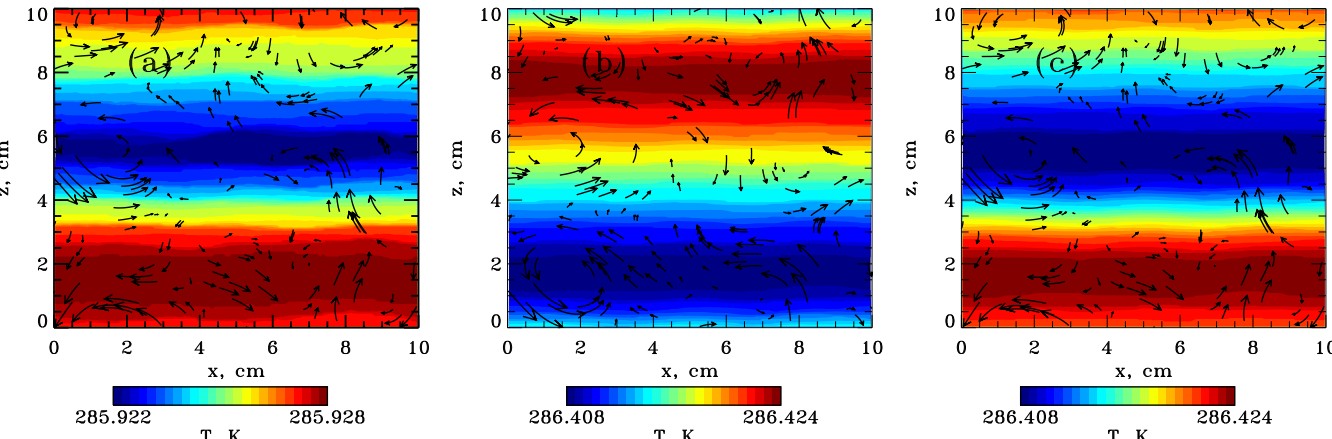

**Fig. 2.** Temperature distribution at $t = 3$ s in *case 1* (a), *case 2* (b) and *case 3* (c), see Table 1. *Non-equilibrium case with low intensive turbulence* is considered (see Table 2). Velocity vector is shown by arrows (averaged vertical velocity is subtracted).

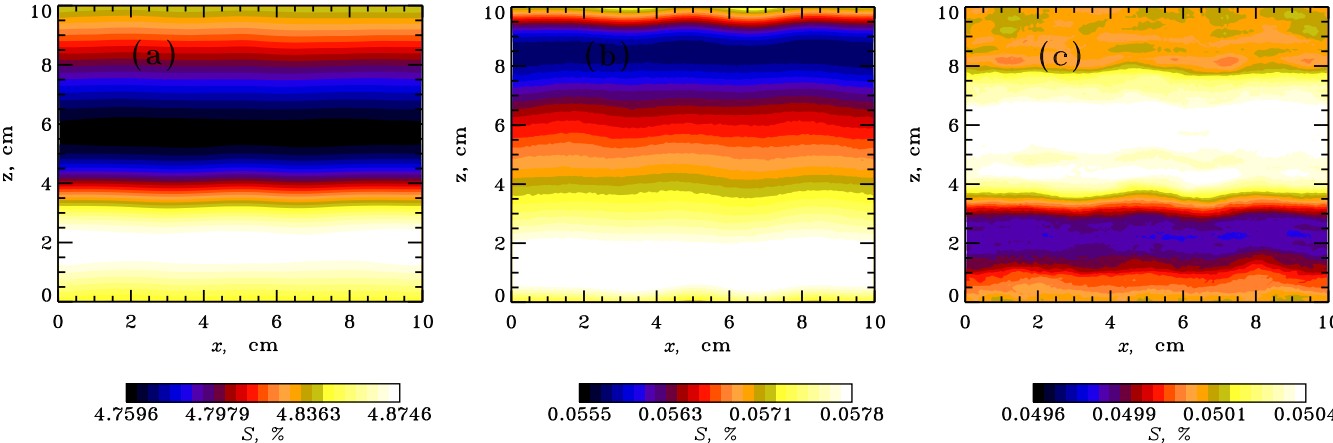

**Fig. 3.** Distribution of supersaturation at $t = 3$ s in *case 1* (a), *case 2* (b) and *case 3* (c), see Table 1. *Non-equilibrium case with low intensive turbulence* is considered (see Table 2).

tial value for mass fraction of water vapor, $Y_{H_2O}$. Oxygen mass fraction is recalculated from the normalization conditions, i.e. $Y_{O_2} + Y_{H_2O} + Y_{N_2} = 1$, where $Y_{O_2}$ is oxygen mass fraction.

*Initial conditions*

This model represents the 3D fluid flow on the microscale inside a volume of 10 cm x 10 cm x 10 cm, just inside the cloud in the mid-troposphere. Based on data of CARRIBA observations typical for the upper parts of clouds / cloud edges at a height of 2000 m, we set the initial conditions for air temperature ($T_0 = 285.4$ K) and water vapor mixing ratio ($q_0 = 0.0124$). The small vertical gradients of temperature and water content are also based on the CARRIBA measurements: the total difference between values of air temperature and water vapor mixing ratio at the upper and lower edges of

the domain are $\Delta T = 0.001$ K and $\Delta q = 4 \times 10^{-5}$, correspondingly.

Observed temperature and absolute humidity result in peak value of supersaturation ($S$) of up to 0.1 (e.g. 10 %). Such values of $S$ are extremely high considering only an adiabatic lifting of a cloudy air parcel. This "quasi-steady state" supersaturation depends mainly on vertical updraft velocity, droplet number concentration, and mean droplet diameter and appears to be on the order of a few tenths of percent for vertical updrafts of about 1 m s$^{-1}$. However, it is well known that atmospheric clouds are mainly non-adiabatic due to turbulent mixing, and a few percent supersaturation are realistic for higher updraft velocities (e.g., Korolev and Mazin, 2003). This is particular true at cloud edges where entrainment of unsaturated air into the cloud results in strong mixing and fluctuations of the water vapor and temperature field. It strongly depends on the correlation between these two ther-

modynamic fields whether strong fluctuations can result in high fluctuations of supersaturation or not. Until now we are not aware of any observations of this correlation in clouds. Based on theoretical arguments and observations in the convective boundary layer below clouds Kulmala et al. (1997) provided some convincing arguments that at the cloud base high fluctuations of supersaturation on the order of up to several percent can exist. It is also well known that turbulence in high Reynolds number flows (typical in convective clouds) is highly intermittent (Siebert et al., 2010). Shaw (2000) argued that under such conditions long-living vortex tubes could produce small areas with decreased droplet number concentrations and, therefore, high supersaturation resulting in secondary activation. Summarizing, there are good arguments that supersaturation fluctuations of a few percent can be generated without strong updrafts.

On the other hand, the supersaturation excess would be eliminated by condensation onto droplets and quasi-steady state supersaturation would be restored (Korolev and Mazin, 2003). Therefore, the key question about the temporal time scale is under consideration now. The ratio of two time scales is important here: the phase relaxation time which describes how fast the supersaturation can react on the new thermodynamic condition and the turbulent mixing time scale which describes how fast turbulence can mix a certain volume (eddy with the length scale $l$). If the phase relaxation time is smaller than the turbulent mixing time then the actual supersaturation will tend to the quasi steady-state solution. However, for scales where the turbulent mixing is faster we expect strong supersaturation fluctuations to "survive".

Let us now assume a small eddy of size $l = 1$ m and a local energy dissipation rate of $\epsilon = 0.1 \mathrm{m}^2\mathrm{s}^{-3}$. These dissipation values are typical peak values for cumulus clouds on that small scales. The highest dissipation can be found at Kolmogorov size (see discussion in Siebert et al., 2006, 2010). The eddy turn-over time is $\tau_{eddy} = (l^2/\epsilon)^{1/3} \approx 2$ s. If we now take a phase relaxation time of the order of one second which is typical for cumulus clouds (see again Korolev and Mazin (2003)) we see that these two time scales are of the same order. Therefore, we conclude that on scales below one meter, strong supersaturation fluctuations can exist and the quasi-steady state solution should be considered as a mean value with superimposed fluctuations with amplitudes up to several percent. This argumentation partly follows the discussion in Section 8e by Korolev and Mazin (2003).

Kulmala et al. (1997) estimated standard deviations of supersaturation of up to 5 % based on aircraft observations at cloud level but outside the clouds. Ditas et al. (2012) observed supersaturation fluctuations in a field of stratocumulus clouds and estimated from highly collocated temperature and water vapor observations peak-to-peak values of up to 1.5 %, which is much higher compared to the quasi-steady-state solution. It is straightforward to assume that higher $S$ values can be expected for parts of (shallow) cumulus clouds. Thus, we argue that for our small modeled volume a transient supersaturation of 10 % can be realistic for specific mixing event.

The pressure, $p$, is assumed to be constant everywhere in the domain and it is also based on measurements. The air density, $\rho$, is calculated from the equation of state $p = \rho R T / m$, where $R$ is gas universal constant, $m$ is air molar mass. The initial velocity is taken to be zero.

To generate the initial turbulent field we make first 100 iterations without evaporation/activation of aerosol particles (further "aerosol dynamics"), including randomly directed external forces (see next section). After that the external forces are set to zero, whereas the particles start to evolve. Thus, the turbulence is decaying for the analyzed time. The maximal time step allowed by the Courant condition for convergence is $\Delta t_c = 10^{-6}$ s (Courant time step). Since $\Delta t_c$ is much larger than the time step needed to move the smallest particle to the neighbor size bin ($\Delta t_a = 2 \times 10^{-7}$ s), at every Courant time step we make 5 sub steps and calculate the particle evolution equation only.

*Boundary conditions*

In all three directions we set periodic boundary conditions for all variables, including the number density function. It means, that at every time step $\Delta t_c$ the number of particles appearing on the bottom/left boundaries is equal to the number of particles disappearing through the top/right boundaries. While the periodic boundary conditions modify the initial temperature stratification, they allow us to consider this domain as an isolated volume, i.e. the total mass, energy and number of particles in the domain does not change with time. In turn, it makes possible to compare the results of simulations, varying the key parameters of the model and to carry out the detailed quantitative analysis of the interaction between aerosol and turbulence.

*Basic equations*

The detailed description of the main equations is presented by Babkovskaia et al. (2015). We consider the standard compressible Navier-Stokes system including equation for conservation of mass, momentum, energy and chemical species. The momentum of air is transported due to viscous force. To describe the gravitational effect we add the buoyant force $B$ to vertical component of momentum equation as

$$B = g \left[ \frac{T - T_0}{T_0} + \epsilon(Y_{H_2O} - q_0) - q_c \right], \tag{1}$$

where $g = 9.81$ m s$^{-2}$ is the acceleration of gravity, $q_0$ is the reference water vapor mixing ratio, $\epsilon + 1 = R_v/R_d$ is the ratio of the gas constant for water vapor and dry air, and $q_c$ is the cloud water mixing ratio (Andrejczuk et al., 2004; Babkovskaia et al., 2015). Note, that the buoyancy force is applied to the air vertical acceleration/deceleration. The air temperature can evolve due to thermal diffusion,

viscous heating, adiabatic contraction/expansion and latent heating/cooling caused by condensation/evaporation on the droplet surface. Coefficient of thermal conductivity and kinematic viscosity are calculated for a mixture of three air species.

We consider evolution of aerosol number density function happening because of evaporation/condensation of aerosol particles. There is energy exchange between particles and ambient gas due to release/absorb of latent heat caused by condensation/evaporation on the droplet surface. The motion of the particles is determined exclusively by their involvement in the motion of surroundings. In the present study we modify water vapor pressure over a droplet of radius $r$ as (Seinfeld and Pandis, 2006; Sorjamaa and Laaksonen, 2007)

$$p_{vs} = p_0 \exp\left(\frac{A}{2r} - \frac{0.1d_w}{2(r - r_0)}\right) \tag{2}$$

where $p_0$ is water vapor pressure over a flat surface at the same temperature, $A = 0.66/T$ (in $\mu$m), where $r_0 = 10$ nm is the radius of the droplet core, $d_w = 0.3$ nm is the size of water molecule.

For generation of the initial turbulence the external forcing $f$ is used in a form

$$f(\mathbf{x}, t) = Re\{N f_k(t) \exp[i\mathbf{k}(t) \cdot \mathbf{x} + i\phi(t)]\}, \tag{3}$$

where $\mathbf{x}$ is the position vector. The wave vector $k(t)$ and the random phase $-\pi < \phi(t) \le \pi$ change at every time step; $N = f_0 c_s(|k|c_s/\Delta t_c)^{1/2}$ is the normalization factor, $c_s$ is the sound speed, $f_0$ is a non-dimensional forcing amplitude; $f_k = (\mathbf{k} \times \mathbf{e})/\sqrt{\mathbf{k}^2 - (\mathbf{k} \cdot \mathbf{e})^2}$, where $\mathbf{e}$ is an arbitrary unit vector that is real and not aligned with $\mathbf{k}$ (see detail in Pencil Code (2001)).

To study the effect of aerosol on the turbulence we consider two initial turbulent fields, taking different non-dimensional forcing amplitudes as $f_0 = 10$ (*low intensive turbulence*) and $f_0 = 100$ (*high intensive turbulence*). Also, we compare the effect of the aerosol in the case when the domain is in equilibrium with environment (*equilibrium* case) and when the temperatures inside and outside the domain are different (*non-equilibrium* case), taking the environment temperature $T_0 = 285.4$ K for *equilibrium case* and $T_0 = 283.5$ K for *non-equilibrium case* . The summary of four considered cases is in Table 2. To study the effect of total number of particles on activation, air temperature and supersaturation we consider *non-equilibrium case* with *low intensive turbulence*.

## 3 Results

*Effect of total number of particles on air temperature and supersaturation distributions*

In Fig. 2 we present the air temperature averaged in $y$-direction when the aerosol dynamic is included. We find that

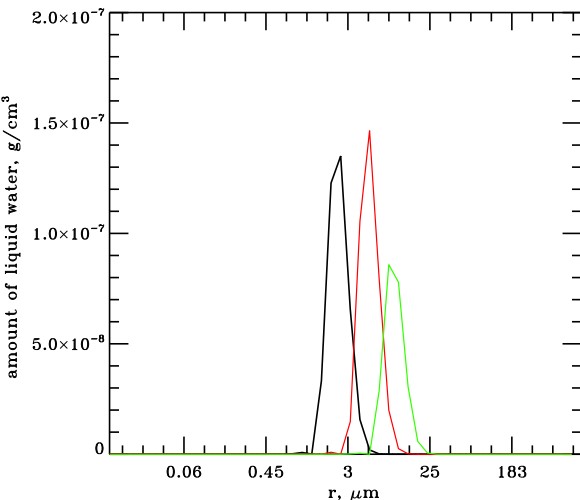

**Fig. 4.** Amount of liquid water accumulated in particles with corresponding radius $r$ at $t = 3$ s (*solid curve* ) for the cases of $N_{tot} = 55.5$ cm$^{-3}$ (*green solid curve*), $N_{tot} = 555$ cm$^{-3}$ (*red solid curve*), $N_{tot} = 5550$ cm$^{-3}$ (*black solid curve*).

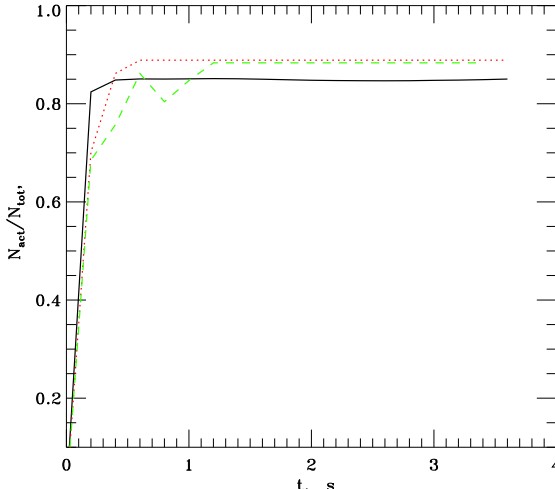

**Fig. 5.** Relative number of activated particles as a function of time $N_{tot} = 55.5$ cm$^{-3}$ (*green curve*), $N_{tot} = 555$ cm$^{-3}$ (*red curve*), $N_{tot} = 5550$ cm$^{-3}$ (*black curve*).

|  | case 1 | case 2 | case 3 | case 4 |
|---|---|---|---|---|
| $\overline{S_{init}}, \%$ | 10.3 | 10.3 | 10.3 | 0.6 |
| $N_{tot}, \mathrm{cm}^{-3}$ | 55.5 | 555 | 5550 | 5550 |
| $N_{act}, \mathrm{cm}^{-3}$ | 49 | 493 | 4700 | 395 |
| LWC, g/m$^3$ | 0.23 | 0.369 | 0.376 | 0.0377 |
| $\Delta$ T, K | 0.52 | 1.0 | 1.0 | 0.05 |
| $\Delta S_{max}, \%$ | -5.42 | -10.11 | -10.47 | -0.69 |
| $\Delta S_{min}, \%$ | -5.66 | -10.12 | -10.48 | -0.37 |
| $\tau_{phase}$, s | 4 | 0.77 | 0.17 | 0.6 |

**Table 3.** Initial value of supersaturation averaged over the domain $\overline{S_{init}}$, total number of particles ($N_{tot}$), number of activated particles ($N_{act}$) at $t = 3$ s, liquid water content (LWC) at $t = 3$ s, change in temperature between start and end of simulation ($\Delta$ T), change in percentage maximal supersaturation between start and end of simulation ($\Delta S_{max}$), change in percentage minimal supersaturation between start and end of simulation ($\Delta S_{min}$), the phase relaxation time of supersaturation $\tau_{phase}$ at $t = 3$ s for considered *cases 1, 2, 3, 4* (see Table 1). The phase relaxation time $\tau_{phase}$ is the numerically predicted value in *cases 2, 3, 4*. In *case 1* the phase relaxation time is obtained from Eq. 6. No subsaturation is predicted anywhere in the domain in all cases. *Non-equilibrium* case with *low intensive turbulence* is considered (see Table 2).

in *case 3* (see Fig. 2, c) after 3 s the difference between the absolute value of maximal and minimal temperatures in the domain is about 0.02 K, whereas without aerosol (*case 5,* not shown) after 3 s the temperature difference is still about 0.001 K. Moreover, the temperature increases by about 1 K everywhere in the domain because of condensation of water vapor on aerosol particles.

Changing of temperature distribution with time is mostly attributable to the periodic boundary conditions: the coldest layers are moving from the bottom to the middle of the domain. However, comparing (a), (b) and (c) panels in Figs. 2 and Fig. 3 we note that the positions of the coldest layers in different panels are different. The coldest layers in case of $N_{tot} = 55$ cm$^{-3}$ coincide with the layers where supersaturation is minimal (see panel a), whereas for other two cases, $N_{tot} = 555$ cm$^{-3}$ (b) and 5550 cm$^{-3}$ (c), the coldest layers correspond to the positions where supersaturation achieves maximum. It happens because of the different effect of aerosol dynamics in dependence on $N_{tot}$. The small amount of aerosol particles (*case 1*) does not make any substantial effect on the temperature distribution, i.e. both temperature and supersaturation are identically shifting with time in vertical direction. In turn, in *cases 2* and *3* aerosol dynamics crucially changes the temperature distribution. One can see in Figs. 2, 3 that layers with larger temperature correspond to layers with smaller supersaturation. It can be interpreted as follows. More intensive condensation occurs in initially warmer layers because supersaturation is larger there and respectively the temperature increases faster in these layers. In turn, since supersaturation exponentially depends on temperature, $S \propto \exp(-T)$, at some moment $S$ appears to be

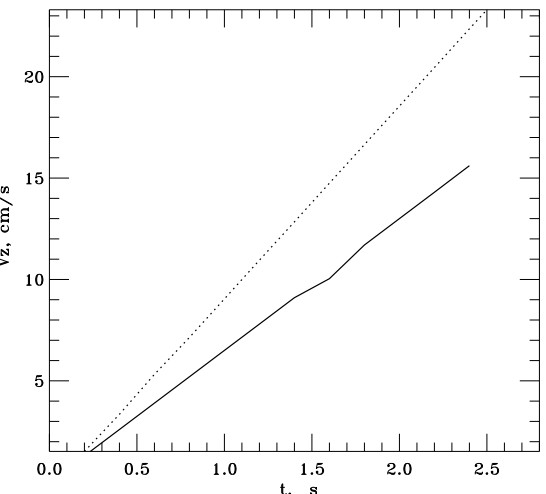

**Fig. 6.** Averaged vertical velocity as a function of time with aerosol for $N_{tot} = 5500$ cm$^{-3}$ (*dotted curve, case 3*); and without aerosol (*solid curve, case 5*). *Non-equilibrium* case with *low intensive turbulence* is considered (see Table 3).

smaller in warmer than cooler layers. Thus, equilibrium supersaturation is higher in the layers with temperature minimum (and vice versa). Also, we find that in *case 1* at $t = 3$ s the supersaturation is about 5 % and the aerosol is still activating, whereas in *case 3* for the first 3 s the supersaturation almost drops to zero and the system is coming to an equilibrium.

Additionally, we investigate how fast the system with initially high value of supersaturation ($S = 10$ %) comes to an equilibrium ($S \simeq 0$ %) in dependence on different total numbers of droplets to answer the question how $N_{tot}$ affects the phase relaxation time. Analyzing Fig. 3 we find that in a case of $N_{tot} = 55$ cm$^{-3}$ it takes more than 3 s for the system to come to equilibrium, i.e. phase relaxation time appears to be larger than theoretically estimated turbulent mixing time ($\tau_{eddy} = 2$ s) (see discussion in section 2). Thus, strong supersaturation fluctuations can "survive" longer if the total number of droplets is small.

*Effect of total number of particles on activation*

In this study we consider that all particles with radius larger than $r_{cr} = 1.75$ $\mu$m are activated. This value is somewhat arbitrary but the results of our study were not sensitive to the choice of $r_{cr}$ provided that $r_{cr} \le 1.75$ $\mu$m. In Fig. 1 we present the normalized particle distribution at $t = 0$ s and $t = 3$ s. We find that the smaller the total number the larger particles are produced for the similar time period. Comparing the supersaturation at $t = 3$ s in these three cases (Fig. 3) one can see

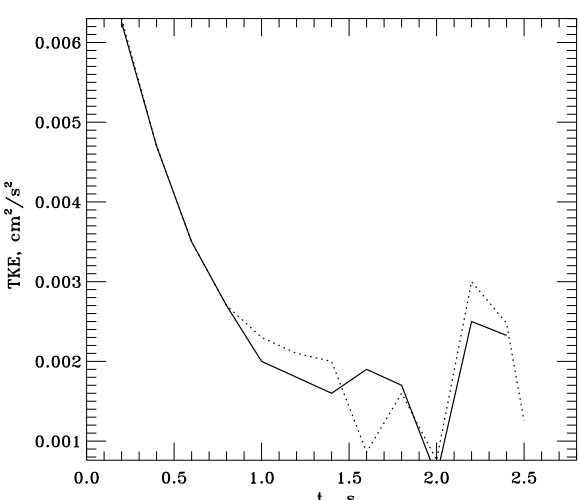

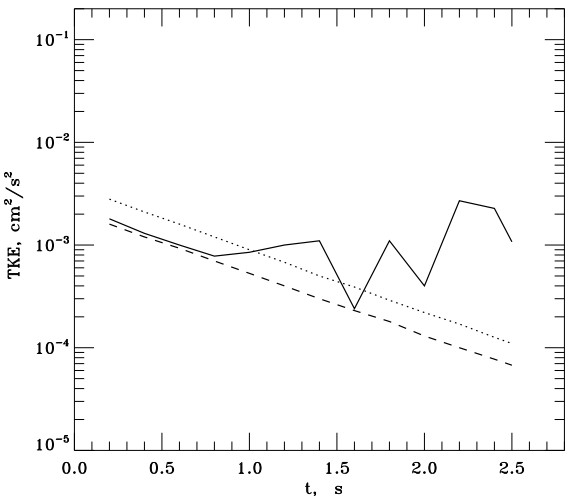

**Fig. 7.** The dependence of the average turbulent kinetic energy on time with aerosol for $N_{tot} = 5500$ cm$^{-3}$ (*dotted curve, case 3*); and without aerosol (*solid curve, case 5*). The other parameters correspond to *non-equilibrium* case with *low intensive turbulence* (see Table 2).

**Fig. 9.** The dependence of the *x*-component (*dotted curve*), *y*-component (*dashed curve*) and *z*-component (*solid curve*) of the averaged kinetic energy on time with aerosol for $N_{tot} = 5500$ cm$^{-3}$.

that in the case of the largest $N_{tot}$ the supersaturation appears to be close to zero and particles stop to grow, while for the smaller $N_{tot}$ the supersaturation is about 4.8 % and particles continue growing. Therefore, the system is coming to an equilibrium (i.e. $S \simeq 0$) faster and aerosol particles cease to grow earlier for the larger $N_{tot}$. Indeed, the phase relaxation time can be estimated as

$$\tau_{phase} = (a_2 N_{tot} \bar{r})^{-1}, \tag{4}$$

where $\bar{r}$ is the mean droplet radius and $a_2 = 3.04\ 10^{-4}$ m$^2$/s. The steady-state supersaturation can be written as $S_{qs} \propto a_1 w \tau_{phase}$, where $a_1$ is a parameter including thermodynamic parameters and being almost constant, $w$ is vertical velocity. Thus, it becomes clear that for larger $N_{tot}$ the system comes to an equilibrium faster. We show that aerosol particles can grow and possibly achieve the size of the rain drop only in the case of small total number of particles. This is consistent with the fact that in natural clouds, deep ascent is needed to drive the prolonged condensational growth of drops in order to form rain (e.g. Rogers and Yau 1989, a short course in cloud physics). In real clouds, the ascent is the source of larger-scale supersaturation. One should also mention, that in the scope of this model we neglect collisions and coalescence of aerosol particles (crucial in creation of rain drops) because of the short total simulation time. Also, due to the short simulation time the droplets are too small for an effective coalescence process.

In addition, in Fig. 4 we analyze the amount of liquid water, $LW(r) = (4/3)\pi r^3 \rho_w N(r)$, where $N(r)$ is the number of particles with radius $r$, and $\rho_w$ is liquid water density. In

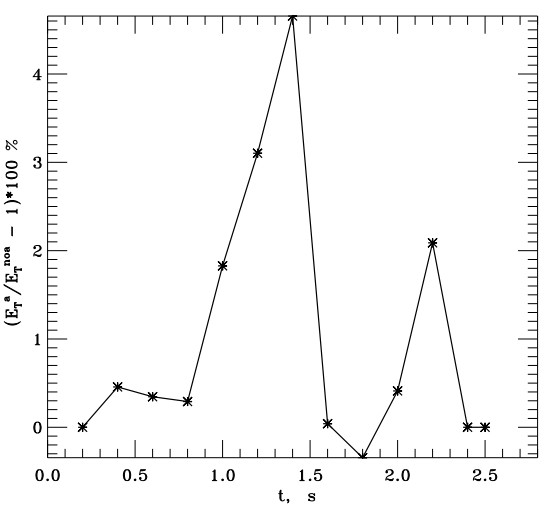

**Fig. 8.** The difference between turbulent kinetic energy averaged over time period $t$ in a case without aerosol ($E_T^{noa}$) and with aerosols ($E_T^a$), where $E_T(t) = 1/t \int_0^t TKE(t')dt'$. All other parameters are the same as in Fig.7.

Fig. 5 we show the fraction of activated particles averaged over the domain as a function of time in *cases 1, 2, 3*. In Table 3 we collect the most important quantities, such as initial value of supersaturation ($\overline{S_{init}}$), number of activated particles ($N_{act}$) at $t = 3$ s, liquid water content $LWC = \int LW(r)/\Delta r\, dr$ (where $\Delta r$ is a size of corresponding bin), change in temperature between start and end of simulation ($\Delta$ T), change in percentage maximal supersaturation between start and end of simulation ($\Delta$ $S_{max}$), change in percentage minimal supersaturation between start and end of simulation ($\Delta$ $S_{min}$), the phase relaxation time of supersaturation $\tau_{phase}$ for *cases 1, 2, 3, 4* (see Table 1). Note that no subsaturation is predicted anywhere in the domain in all considered cases.

We find that while the total number in *case 2* is ten times smaller than in *case 3*, liquid water content is similar in these two cases. On the other hand, *LWC* appears to be smaller in *case 1* than in *cases 2, 3* (see Table 3 and Fig. 4). It happens because the probability of water molecules to catch a particle is much smaller in a case of the smallest particle concentration (*case 1*) than in *cases 2, 3*. In Fig. 5 one can see that in *case 1* and *case 2* the number of activated particles does not grow after 1.2 s and 0.6 s, correspondingly, and in *case 3* it happens after 0.5 s. Since in *case 3* the equilibrium is achieved earlier than in *cases 1* and *2*, the maximum of final particle distribution in *case 3* is shifted (see Fig. 1) and the final relative number of activated particles appears to be smaller than in *cases 1* and *2*. We find that at t= 3 s the number of activated particles is proportional to the total number, whereas the change of $N_{tot}$ by a factor of hundred increases *LWC* by approximately 40 % (Table 3).

*Effect of aerosol on the turbulent motion*

We analyze the effect of aerosol on the turbulent motion, taking $N_{tot} = 5500$ cm$^{-3}$ (*case 3* in Table 1). First, we consider the *equilibrium case*, taking $T_0 = 285.4$ K and $q_0 = 0.0124$ in Eq. 1. In that case the turbulent field appears to be isotropic. Next, we decrease $T_0 = 283.5$ K and study the developing of turbulence in *non-equilibrium case*. In that case the intensive vertical motion is generated due to buoyant force. Note, that the model domain is not vertically displaced during the simulation time but the vertical motion is generated within domain. Also, we vary parameter $f_0$ in Eq. 3 to compare the developing of *low intensive* ($f_0 = 10$) and *high intensive* ($f_0 = 100$) turbulence in both *equilibrium* and *non-equilibrium* cases. The summary of parameters is presented in Table 2. Finally, we compare the results of simulations with and without particles in all four cases described above.

We find that the vertical motion of air is accelerated because of aerosol dynamics. Also, in Fig. 8 we show the time averaged turbulent kinetic energy, $E_T(t) = 1/t \int_0^t TKE(t')dt'$, as a function of time in the case with aerosol particles and without them. We conclude that turbulent kinetic energy increases because of presence of aerosols. We interpret these results as follows. The air temperature increases because of

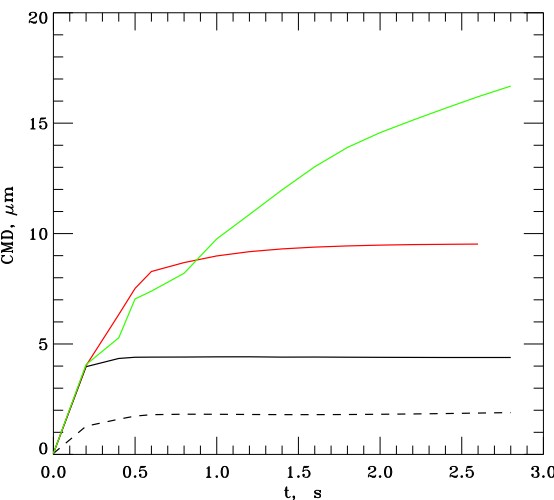

**Fig. 10.** The dependence of count mean diameter (CMD) on time for *case 1* (*green solid curve*), *case 2* (*red solid curve*), *case 3* (*black solid curve*), *case 4* (*black dashed curve*).

release of latent heat caused by condensation onto droplets, and therefore, the difference between temperatures inside and outside the domain is enlarged. It results in increasing of buoyant force and accelerating of air motion in vertical direction.

Also, we find that acceleration does not depend on intensity of turbulent fluctuations, i.e. acceleration in vertical direction is the same in *low intensive turbulence* and in *high intensive turbulence* cases. Moreover, turbulent fluctuations grow because of presence of aerosol particles in all four considered cases. The dependences of vertical velocity and turbulent kinetic energy (TKE) averaged over the domain as a function of time for *non-equilibrium* case with *low intensive turbulence* are presented in Figs. 6 and 7, correspondingly.

Finally, we find that there is a strong variation of TKE with time (see Fig. 7). To interpret this fact we plot *x*- ,*y*- and *z*- components of TKE in Fig. 9 and find strong time variations only in *z*-component of TKE (*x*- and *y*- components are smoothly decreasing with time). We conclude that it results from fluctuations of temperature caused by aerosol dynamics, and therefore, changes of buoyant force with time, which result in perturbation of vertical motion.

*Effect of initial supersaturation on activation of aerosol particles*

To illustrate the effect of initial supersaturation on activation of aerosol particles we modify the initial distribution of absolute humidity in *case 3* decreasing it by 10 %. In that case the initial supersaturation averaged over domain becomes 0.6

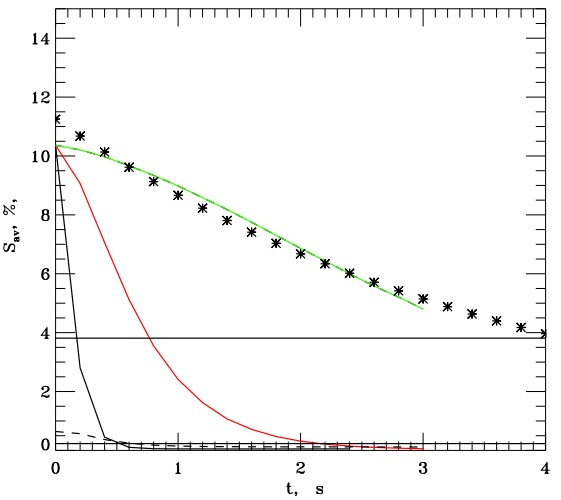

**Fig. 11.** The dependence of supersaturation averaged over the calculation domain on time for *case 1* (*green solid curve*), *case 2* (*red solid curve*), *case 3* (*black solid curve*), *case 4* (*black dashed curve*). Horizontal lines show the corresponding values of $\overline{S_{init}}/e$. Asterisk show the extrapolation function in *case 1* (see Eq. 6).

%. To study the effect of initial supersaturation on activation of aerosol particles in Fig. 10 we analyze count median diameter (CMD) taking

$$\text{CMD} = (D_1^{n_1} D_2^{n_2} D_3^{n_2} ... D_{N_{tot}}^{n_{N_{tot}}})^{1/N_{tot}}, \tag{5}$$

where $D_i$ is particle size of the $i^{th}$ bin, $n_i$ is number of particles having a size $D_i$. Dependences of CMD on time in *cases 1, 2, 3, 4* are presented in Fig. 10. Comparing *cases 3* and *4* we find that in both cases the system comes to equilibrium at $t \simeq 0.6$ s. In other words, the typical time it takes for droplets to cease growing does not depend on initial supersaturation. On the other hand, a value of CMD in equilibrium crucially depends on supersaturation and it appears to be two times larger in *case 3* than in *case 4*.

Additionally, in Fig. 11 we present the supersaturation averaged over domain for *cases 1, 2, 3, 4* (see Table 1) and analyze the phase relaxation time of supersaturation ($\tau_{phase}$) for different values of initial supersaturation and total number of particles. Analyzing numerical results we get the phase relaxation time of about 0.77 s (*case 2*), 0.17 s (*case 3*) and 0.6 s (*case 4*). Using Eq. 4 and count median diameter (see Fig. 10) we estimate the phase relaxation time of 1.23 s (*case 2*), 0.27 s (*case 3*) and 0.63 s (*case 4*). Therefore, we conclude that results of our simulations are in a good agreement with analytical estimations. In *case 1* the phase relaxation time is larger than time of simulations. In that case we extrapolate the numerical results with the following function

(see asterisk in Fig. 11)

$$S_{av}(t) = 11.25 \exp\left(-\frac{t}{3.84}\right) \tag{6}$$

and estimate the phase relaxation time of about 4 s. The results are summarized in Table 3. We conclude that $\tau_{phase}$ depends both on initial supersaturation and on the total number of particles.

## 4   Summary

Turbulence, aerosol growth and microphysics of hydrometeors in clouds are intimately coupled. In the present study a new modeling approach was applied so as to quantify this linkage. We study the interaction in the cloud area under transient, high supersaturation conditions, using direct numerical simulations. As the initial conditions we take observational data. To analyze the effect of aerosol and droplets on turbulence a small volume with supersaturation of 10% was considered. Under such extreme conditions, condensation is the dominant process. The results cannot be linearly extended to bigger cloud volumes but should be considered as relevant for a small cloud parcel with extreme supersaturation due to turbulent mixing of the water vapor and temperature field. As an initial distribution of particles we take the data of measurements at the sea level and analyze the droplets activated by the aerosols in the simulations.

To study the effect of total number of particles on activation, air temperature and supersaturation we vary the total number of particles and take the other parameters corresponded to *low intensive turbulence* and *non-equilibrium case*, i.e. when the vertical motion is generated within domain because of buoyancy. We compare the results of simulations with particles and without them. We find that the total number of particles in the domain is crucial for distribution of temperature and for developing of turbulence. Even small amount of aerosol particles (55.5 cm$^{-3}$) and therefore small cloud droplet number concentrations increase the air temperature by 1 K because of latent heating caused by condensation onto drops. The system comes to an equilibrium faster for the larger total number of particles. To illustrate the effect of initial supersaturation on activation of aerosol particles we compare the results of simulations with initial supersaturation averaged over domain of 10.3 % and 0.6 %. We conclude that the typical time it takes for droplets to cease growing does not depend on initial supersaturation. Also, we analyze the phase relaxation time of supersaturation (i.e. time of dropping from initial value to $1/e = 0.368$) for different values of initial supersaturation and total number of particles. We find that the phase relaxation time crucially depends both on the total number of particles and on initial supersaturation.

To analyze the effect of aerosol dynamics on the turbulent kinetic energy and on vertical velocity we take the maximal value of $N_{tot} = 5550$ cm$^{-3}$. We conclude that the presence of aerosol has an effect on vertical motion and in our

case (when the temperature inside the domain is larger than in environment) tends to enlarge upward velocity. We conclude that aerosols quite strongly influences the dynamics in the cloud area. Such effect of aerosols can be crucial also for large scales usually studied with Large Eddy Simulation (LES) and the LES parametrization can be improved with Direct Numerical Simulations.

### Acknowledgments

We thank the Helsinki University Centre for Environment (HENVI) and computational resources from CSC – IT Center for Science Ltd are all gratefully acknowledged. This research is supported by the Academy of Finland Center of Excellence program (project number 272041). Anthropogenic emissions on Clouds and Climate: towards a Holistic Under-Standing" (BACCHUS), project no. 603445.

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
