# Peer review of "A DNS study of aerosol and small-scale cloud turbulence interaction"

_Atmospheric Chemistry and Physics, 2015_

## Referee Comment (RC1) · Anonymous Referee #1 · 15 Feb 2016

**General:**

The study on aerosol particle dynamic effects is a spectacular idea and performance on small scale variation effects on cloud properties such as activated particles and temperature effects usually either ignored or simply parameterized. The approach and the implications for example for larger scale aerosol particle – cloud effect calculations matches nicely in the scope of Atmospheric Chemistry and Physics and the results are quite interesting. However before accepting the present study I would recommend several technical improvements and clarifications in order to support readers not essentially familiar with all the details to follow the arguments and the implications for larger scale simulations. Those include first of all the English. Please have a native English speakers check on the sentences!
[Figure]

**Detailed questions and comments:**

- The total number of particles was varied between two orders of magnitude, which was extracted from reasonable values measured. This is appropriate and reasonable. However, what about the impact of different size ranges e.g. mode concentrations on the results? Do the results change notably for particles in the accumulation and in the coarse mode due to critical sizes for activation for the salt particles assumed? Would results differ for changing certain size bin concentrations (i.e. modes) instead of the whole number? Are these salt particles already "activated" or assumed "dry" for the simulations conducted? I guess once any of these particles has faced substantial humidity it will grow much easier than if it has to dissolve first.

- Abstract, p.1: "The system comes to an equilibrium faster and the relative number of activated particles appears to be smaller for larger $N_{tot}$." seems to be formulated very simple. I doubt that for a large part of atmospheric processes equilibrium conditions are hardly reached. What is the criteria for achieving an equilibrium condition in this case and for which simulation conditions the equilibrium approach becomes invalid?

- Description of the model, p. 2: The order of figures seems somewhat arbitrary, as Fig. 3 appears earlier than Fig. 1.

- p. 2, l. 106ff: The particle size distribution displays a sharp maximum close to a diameter of around 5 micron. Please refer to the origin of observations (reference, location etc.) mentioned in the text.
- p. 2, l. 127: It's being referred to a temperature gradient of 0.001 K. Two questions on that: (i) which gradient, i.e. temperature change over which distance, horizontal, vertical etc.? Only a temperature unit is provided. (ii) This temperature change is pretty tiny although important. What is the reliability range of this because of numerical diffusion and linearization of equations for simulation? Please provide a temperature gradient and either a short statement of simulation uncertainty or a $\pm$ value.

- p. 3, Fig. 2: I do understand the intention to maximize differences in the colour scale to make aspects visible. However, since in here three situations are compared with, please use the same scale for all the three upper and all the three lower plots. This would allow a better comparison and an even improved identification of the changes.

- p. 3, l. 163ff: Please reformulate: "... and the usual equilibrium supersaturation would be restored.". I doubt an equilibrium supersaturation, as water tends to equilibrate at saturation. If you mean different, please reformulate to make it clearer.

- p. 3, l. 172ff: Please check: "If the phase relaxation ... would be applicable." There seem to be too many words. Is the word "than" dispensible?

- p. 4, l. 229: You state that the number of particles stays constant. This contradicts the explanation of an aerosol particle dynamic study. Are changes if calculated in the corresponding simulation time negligible? Otherwise this may matter as e.g. larger cloud droplets grow on the expense of smaller droplets and

they modify the size spectrum and number density.

- p. 4, l. 266 and p. 2, Table 2: The change in temperature between equilibrium and unequilibrium case seems fairly huge! 8K would cause a strong vertical uplift, a strong local mixing (dilution), which would require a remarkable mass of condensed water vapour (several grams per m$^3$). Did I get something wrong?

- p. 4, l. 278f: The temperature is averaged in y-direction. If you have notable differences in x- and z-direction, how does this assumption affect the results? To a negligible extend?

- p. 5, Table 3: I don't understand the listed maximal and minimal values of supersaturation S as they are negative. This would imply a subsaturation as S = 1-p/psat0 with p and psat0 the vapour pressures of water at present and at saturation level. Second, very interesting is the change between cases 1 and 2. There seems to be a tipping point at a certain total particle number concentation. Could you provide a comment on that as the changes by a factor of ten is substantial?

- p. 6, l.298f: You state that the simulated results occur because of the effect of total number concentration. Why? I guess a certain limit of aerosol particles – here all assumed to be identical in chemical composition and water solubility – exists, below which the time of diffusion of water vapour to the next aerosol particle is too long to achieve the same amount of condensation. Because of the particles size (predominantly beyond 1 micron in diameter) hardly any curvature effects on saturation vapour pressure can be expected. If so, could you name
the cutting point for the conditions simulated in here?

- p. 6, l. 330: The point mentioned above feeds back to the statement dealing with the activation radius assumed. Why exactly 1.75 micron? This should depend on supersaturation. "…the results of this study were not sensitive on the choice of " the activation radius. My guess (!) is that this is valid for the cases 2 and 3 but not for case 1. Do you agree or disagree?

- Fig. 6: The calculated vertical velocities of 0.6 to 0.7 m/s at maximum are remarkable. It is indicated that this intensifies over time although a steady-state or "equilibrium" is to be achieved after a second or somewhat more.

- p. 6, Fig., 7: "The dependence of the average..." turbulent "kinetic energy…". Please insert.

- p.7, 353f: Aerosol dynamics are neglected. This sounds different in the abstract as it is stated that in order "to study effects of aerosol dynamics on the turbulence we vary…". Please name explicitly in the methods section not to use aerosol dynamics and state that this is valid because of the short total simulation time used.

- p. 7, l. 380ff: "We find that the number … linearly depends…." Please check the English and be careful when using three simulations only. Especially Table 3 (p. 5) contradicts. Better skip that sentence or perform more simulations in more narrow Ntot steps.
- p. 7, l. 400f: "We find that the vertical motion of air is decelerated because of aerosol dynamics." This contradicts to the statement of neglecting aerosol dynamics (condensation and coagulation) during the period of simulation (p. 7, l. 353)! Please check.

- p. 8, l. 405ff: You explain the air temperature change driven by the condensation of water vapour onto the aerosol particles and the release of latent heat. But since the aerosol particles are rather huge size shouldn't matter and the condensation should occur independent on the number if any particle number and time are available. But the change differs notably between 55 and 550 cm$^{-3}$ and I can only think of not sufficient time for condensation.

- p. 8, l. 436f. The information on the model sizes is very nice but would be best to include it earlier in the methods section for a better understanding on set-up and interpretation of results.

- p. 8, l. 450ff: Very nice indeed. But simulating a 10x10x10 cm$^3$ volume this would cause dramatic horizontal and vertical gradients and motion. Is this still applicable by the present method including the problematic areas along the edges of the finite volume?

- p. 8, end: Very nice and interesting results indeed. I would recommend a short statement to potential implications for cloud simulations and weather prognosis. This would definitely increase the range of potential readers, for which the area is highly relevant.

---

## Referee Comment (RC2) · Anonymous Referee #2 · 26 Feb 2016

This article studies the effect of aerosol dynamics on atmospheric small-scale turbulence using direct numerical simulations.

As I already pointed out in my original assessment of the article, my main concern with this article are the extreme initial conditions that were chosen for the simulations: While I understand the concept of fluctuations and the concurrent possibility of achieving extreme values, it is very hard for me to assess how relevant it is to study such an extreme case outside of that context. To elaborate on what I mean, let's take the article by Kulmala et al. that has also been cited by the authors: Kulmala et al. treat the saturation ratio (let's call it $S'$ here, because $S$ is used for the supersaturation in the present article) as a stochastical variable with a Gaussian distribution around an average value which varies from 0.995 to 1.0 with a standard deviation of up to 0.05. They then conduct a series of simulations where they allow the saturation ratio to vary

randomly according to the assumed distribution and find that particles can activate also in under-saturated conditions due to the temporal fluctuations in the saturation ratio. In the present paper, the authors pick one very extreme case out of this distribution, which corresponds to a saturation ratio of 1.1 or a supersaturation of 10 % (I can only guess that they still assume the average $S'$ in the cloud to be equal to one). Just to put this into context, common supersaturation values at the base of a cloud are of the order of 0.1 to 0.5 %; according to a quick test conducted with a cloud parcel model that does not consider fluctuations, it reqires a particle number concentration of 1 cm$^{-3}$ and an updraft velocity of 10 m/s to achieve a supersaturation of 10 %. Furthermore the authors chose a very high temperature difference between the simulation domain and its surroundings, which is not motivated in the text at all. According to these extreme initial conditions, the authors then also find that aerosols have a strong influence on turbulence, but I wonder how justifiable such a conclusion is without also considering more moderate supersaturations which are, after all, much more likely to occur. Furthermore, I am a little bit skeptic how reliable the results presented here are, as the simulations include the use of random numbers (this especially concerns the generation of the initial turbulence) and thus a single simulation may not be very representative of an average behaviour.

To conclude, I cannot recommend this article for publication in the current form. At the very least the paper requires one more set of reference simulations with a more conventional supersaturation of, say, 0.3 %, and a proper discussion on the issues I layed out above.

**Concrete remarks**

1. The English is not very good and needs to be reviewed. Some of the sentences are very hard to understand.

2. *lines 12–14:* Latent heat release is a time dependent process, but finally, close to equilibrium, the total cooling depends (nearly) only on the initial amount of supersaturation. It is therefore unnecessary to state that "even small amounts of aerosols increase the air temperature", and it is quite misleading to give the (very high) change in temperature of 1 K without also giving the value for the supersaturation used.

3. *lines 115–120:* Why is the chemical composition of the air important to this study? Wouldn't it be enough to state the total water content?

4. *lines 126–129:* Is this the total difference in temperature/water vapor content between the (upper and lower?) edges of the domain (e.g. 0.001 K/10 cm)? According to the units it cannot be a gradient.

5. *lines 133–135:* How exactly was this range of supersaturation derived?

6. *line 264:* I have never heard the word "unequilibrium" before – I believe the correct term is "non-equilibrium".

7. *Figure 1:* The velocity fields in panels (a) and (c) look curiously similar, while they are totally different in panel (b). Did you use exactly the same initial velocity fields in all three cases (or probably a different one for $N_{\text{tot}} = 555$ cm$^{-3}$)? How much does varying the particle concentration actually affect turbulence?

8. *lines 306–317:* How does the initial supersaturation vary? Are these the values that you mention in lines 133–135? If so, it is not obvious how only the temperature is responsible for this variation in supersaturation, as you also set a gradient in water vapour mixing ratio. I think a figure that clearly lays out the initial conditions of the simulations would be very helpful in the overall understandability of this paper.

[Figure]

9. *lines 323–326:* Can you somehow quantify how long it would take the system to equilibrate, "more than 3 s" is pretty vague?

10. *lines 341–347:* As the condensational growth of particles is limited by the total surface area, it is not very surprising that the water vapor mixing ratio decreases faster if more particles are present. Also, as for this reason the supersaturation is always larger for smaller number concentrations, it should be expected that these particles grow larger. I don't know if this really "confirms the Twomey effect" or if it rather confirms that the model works as it should.

11. *line 359:* Replace "number" with "fraction".

12. *Figure 5:* Why does the fraction of activated particles go down after about half a second for $N_{\text{tot}} = 55.5$ cm$^{-3}$?

13. *lines 380–384:* I don't understand the relevance of this statement. As you have the same total water content in all three cases to begin with, the difference in LWC very strongly depends on the time at which you compare the simulations (apart from some small corrections due to temperature and droplet size), because in some simulations the droplets still grow while in others they don't. How would this difference change if you waited 6 instead of 3 seconds?

14. *lines 385–399 and Figure 6:* How exactly does the buoyancy force affect the turbulence in the simulation domain at scales smaller than 10 cm? This is not discussed at all in the manuscript. How valid is the assumption of a temperature difference of 7.6 K between the simulation domain and its surroundings, especially when the cloud parcel is colder than the rest? From Figure 6 I roughly estimate a deceleration of the air parcel of 25 cm s$^{-2}$ (solid black line), so the air parcel drops more than 1 m (more than ten times the domain size) during the 3 s simulation, still the environment temperature never changes. How does this conform with the original statement that the temperature in the cloud is subject to

strong fluctuations? On a side note, if the temperature changes by 1 K, it is not surprising that the buoyancy force changes quite a bit and I wonder if Figure 6 is necessary at all – why not just give a value for $B$ in the different cases? Furthermore, what does the comparison of the two simulations tell us. No aerosols means no cloud – should you rather compare between different aerosol loadings to somehow mimic what could happen at the edge of the cloud?

15. *lines 400–410:* To my eye the TKE in Figure 7 only differs significantly between aerosols and no aerosol during a short period of time around 2 s. Accordingly, Figure 8 shows a strong jump from roughly zero up to 80 % at that point in time and starts dropping off again thereafter. Still, from Figure 8 the authors conclude that aerosols affect turbulence strongly. How sure are you that these results are not sensitive to the specific set of random numbers used for the simulation (i.e. what would happen if you used a different random seed in the setup)?

16. *lines 406–410:* This sentence is very hard to understand. Do you mean to say that the temperature *difference* between the domain and its surroundings *decreases* because the temperature inside the domain *increases*?

---

## Author Comment (AC1) · 20 May 2016

General: The study on aerosol particle dynamic effects is a spectacular idea and performance on small scale variation effects on cloud properties such as activated particles and temperature effects usually either ignored or simply parameterized. The approach and the implications for example for larger scale aerosol particle – cloud effect calculations matches nicely in the scope of Atmospheric Chemistry and Physics and the results are quite interesting. However before accepting the present study I would recommend several technical improvements and clarifications in order to support readers not essentially familiar with all the details to follow the arguments and the implications for larger scale simulations. Those include first of all the English. Please have a native English speakers check on the sentences!

[Figure]

The total number of particles was varied between two orders of magnitude, which was extracted from reasonable values measured. This is appropriate and reasonable. However, what about the impact of different size ranges e.g. mode concentrations on the results? Do the results change notably for particles in the accumulation and in the coarse mode due to critical sizes for activation for the salt particles assumed? Would results differ for changing certain size bin concentrations (i.e. modes) instead of the whole number? Are these salt particles already "activated" or assumed "dry" for the simulations conducted? I guess once any of these particles has faced substantial humidity it will grow much easier than if it has to dissolve first.

These are very interesting questions and we would definitely like to perform more simulations to answer, but one model simulation for 3s takes about 2 weeks at a cluster machine when using 512 cores. So the computational costs are very high for these runs and for this reason it was not possible to perform runs with e.g. different size bin distributions.

Evolution of the particle distribution function was accurately tested in previous study. (Babkovskaia et al. 2015). The minimal size bin and the time step are chosen such that for one time step the particles move to the neighbor bin. We checked that the increasing/decreasing of the bin size and corresponded decreasing/increasing of the time step do not make any effect on the final result. Also, we assume that initially the particles are almost dry, i.e. there is a solid core with very thin water envelope to start the activation.

1) Abstract, p.1: "The system comes to an equilibrium faster and the relative number of activated particles appears to be smaller for larger Ntot." seems to be formulated very simple. I doubt that for a large part of atmospheric processes equilibrium conditions are hardly reached. What is the criteria for achieving an equilibrium condition in this case and for which simulation conditions the equilibrium approach becomes invalid?

Indeed, the phase relaxation time is tauphase $\sim (Ntot <r>)-1$, where $<r>$ is the mean

droplet radius. The steady-state supersaturation can be written as Sqs ∼ a1 tauphase, where a parameter a_1 m-1 is a parameter including thermodynamic parameters and being almost constant. Thus, it becomes clear that for larger Ntot the system comes faster to an equilibrium.

2) Description of the model, p. 2: The order of figures seems somewhat arbitrary, as Fig. 3 appears earlier than Fig. 1.

Fig 3 is moved to the beginning of section 2

3) p. 2, l. 106ff: The particle size distribution displays a sharp maximum close to a diameter of around 5 micron. Please refer to the origin of observations (reference, location etc.) mentioned in the text.

In the caption of Fig 3 it was mentioned that dotted curve corresponds to the observed distribution of aerosol, and solid curves are distributions of droplets. There are no measurements for 5 micro-meter particles.

4) p. 2, l. 127: It's being referred to a temperature gradient of 0.001 K. Two questions on that: (i) which gradient, i.e. temperature change over which distance, horizontal, vertical etc.? Only a temperature unit is provided. (ii) This temperature change is pretty tiny although important. What is the reliability range of this because of numerical diffusion and linearization of equations for simulation? Please provide a temperature gradient and either a short statement of simulation uncertainty or a value.

This part was rewritten:

This model represents the 3D fluid flow on the microscale inside a volume of 10 cm x 10 cm x 10 cm, just inside the cloud in the mid-troposphere. Based on data of CARRIBA observations typical for the upper parts of clouds / cloud edges in a height of 2000 m, we set the initial conditions for air temperature (T0 = 285.4 K) and water vapor mixing ratio (q0= 0.0124). The small vertical gradients of temperature and water content are also based on the CARRIBA measurements: the total difference between values of air

temperature and water vapor mixing ratio at the upper and lower edges of the domain are (Delta T = 0.001 K) and (Delta q= 4 10-5), correspondingly.

5) p. 3, Fig. 2: I do understand the intention to maximize differences in the color scale to make aspects visible. However, since in here three situations are compared with, please use the same scale for all the three upper and all the three lower plots. This would allow a better comparison and an even improved identification of the changes.

The differences between maximal and minimal values in the corresponding cases are much smaller than the difference between plots. Keeping the same max and min for all three cases we could not resolve the distribution of temperature (supersaturation) in one case. We have left the plots in previous form.

6) p. 3, l. 163ff: Please reformulate: ": : : and the usual equilibrium supersaturation would be restored.". I doubt an equilibrium supersaturation, as water tends to equilibrate at saturation. If you mean different, please reformulate to make it clearer.

Reformulated: On the other hand, the supersaturation excess would be eliminated by condensation onto droplets and quasi-steady state supersaturation would be restored.

7) p. 3, l. 172ff: Please check: "If the phase relaxation : : : would be applicable." There seem to be too many words. Is the word "than" dispensible?

Fixed: If the phase relaxation time is smaller than the turbulent mixing time then the actual supersaturation will tend to the quasi steady-state solution.

8) p. 4, l. 229: You state that the number of particles stays constant. This contradicts the explanation of an aerosol particle dynamic study. Are changes if calculated in the corresponding simulation time negligible? Otherwise this may matter as e.g. larger cloud droplets grow on the expense of smaller droplets and they modify the size spectrum and number density.

Here we mean the total (i.e. integrated over all sizes) number of particles in the domain. The periodic boundary conditions means that the particles can not come in/out from

the domain.

9) p. 4, l. 266 and p. 2, Table 2: The change in temperature between equilibrium and unequilibrium case seems fairly huge! 8K would cause a strong vertical uplift, a strong local mixing (dilution), which would require a remarkable mass of condensed water vapor (several grams per m3). Did I get something wrong?

We have prepared results of simulations in a case of smaller difference between the temperature inside the domain and its surroundings (delta T= 2.5 K). Moreover, now the temperature inside the domain is larger than outside the domain and we get updraft instead of downdraft. We agree that new parameters are more realistic in atmosphere. However, our general conclusion concerning the effect of aerosol on turbulence does not change. We showed that the aerosol affects the turbulence through the buoyancy. However, since now temperature difference changes sign the vertical air motion is accelerated (rather than decelerated as it was before) if aerosol particles are present.

10) p. 4, l. 278f: The temperature is averaged in y-direction. If you have notable differences in x- and z-direction, how does this assumption affect the results? To a negligible extend?

The notable gradients of corresponding variables exist only in vertical direction (z). In x- and y- directions the changes are only because of turbulent fluctuations (not so strong). Therefore, averaging in y- direction does not make any crucial effect on the results.

11) p. 5, Table 3: I don't understand the listed maximal and minimal values of supersaturation S as they are negative. This would imply a subsaturation as S = 1-p/psat0 with p and psat0 the vapor pressures of water at present and at saturation level. Second, very interesting is the change between cases 1 and 2. There seems to be a tipping point at a certain total particle number concentration. Could you provide a comment on that as the changes by a factor of ten is substantial?

We introduce supersaturation as S=p/psat-1. It decreases with time. Therefore, the

difference between values at t=3 s and t=0 s is negative. The caption is rewritten: Initial value of supersaturation averaged over the domain \overline{S_{init}}, total number of particles (Ntot), number of activated particles (Nact) at t=3 s, liquid water content (LWC) at t=3 s, change in temperature between start and end of simulation (Delta T), change in percentage maximal supersaturation between start and end of simulation (Delta Smax), change in percentage minimal supersaturation between start and end of simulation (Delta Smin), relaxation time of supersaturation tr at t=3 s for considered cases 1, 2, 3, 4 (see Table1). In cases 2,3,4 tr is the numerically predicted value (time for 63 \% change form start). In case 1 the phase relaxation time is obtained from Eq.6. No subsaturation is predicted anywhere in the domain in all cases.

Concerning the tipping point mentioned we agree with the referee that this would be an interesting point to study further and will also consider it in our next research. However, because of the high computational costs of the simulations we decided to leave this open for the future. Speculative we would argue that at a certain number concentration the activation for a given supersaturation is not anymore limited by the amount of particles, which would mean that at this time the diffusion limitation of water molecules reaching the particles is negligible. However, to prove this statement more runs with different number concentrations and a set of different supersaturation would be required.

12) p. 6, l.298f: You state that the simulated results occur because of the effect of total number concentration. Why? I guess a certain limit of aerosol particles – here all assumed to be identical in chemical composition and water solubility – exists, below which the time of diffusion of water vapor to the next aerosol particle is too long to achieve the same amount of condensation. Because of the particles size (predominantly beyond 1 micron in diameter) hardly any curvature effects on saturation vapor pressure can be expected. If so, could you name the cutting point for the conditions simulated in here?

No at this time we are not able to name the cutting point as the referee mentioned. As already explained above our hypothesis is that there is a diffusion limitation between

the water molecules and the particles for certain amount of aerosol number concentrations, however, the complete study of this interesting phenomena was out of the frame of this manuscript but will be considered in our future research.

13) p. 6, l. 330: The point mentioned above feeds back to the statement dealing with the activation radius assumed. Why exactly 1.75 micron? This should depend on supersaturation. ": : :the results of this study were not sensitive on the choice of " the activation radius. My guess (!) is that this is valid for the cases 2 and 3 but not for case 1. Do you agree or disagree?

For three cases with the same initial supersaturation the results do not depend on critical radius if it is smaller than 1.75 micron. We have prepared additional run for smaller supersaturation and discussed effect of supersaturation in additional section. We find that for smaller supersaturation the system comes to an equilibrium for the same time as in a case of large supersaturation but the final size of particles appear to be smaller. We agree that to generalize our results for different initial parameters more simulations are needed.

14) Fig. 6: The calculated vertical velocities of 0.6 to 0.7 m/s at maximum are remarkable. It is indicated that this intensifies over time although a steady-state or "equilibrium" is to be achieved after a second or somewhat more.

We have prepared results of simulations in a case of smaller difference between the temperature inside the domain and its surroundings (delta T= 2.5 K). Moreover, now the temperature inside the domain is larger than outside the domain and we get updraft instead of downdraft. We agree that new parameters are more realistic in atmosphere. However, our general conclusion concerning the effect of aerosol on turbulence does not change. We showed in the last section that the aerosol affects the turbulence through the buoyancy. However, since now temperature difference changes sign the vertical air motion is accelerated (rather than decelerated as it was before) if aerosol particles are present.

15) p. 6, Fig., 7: "The dependence of the average..." turbulent "kinetic energy: : :". Please insert.

inserted

16) p.7, 353f: Aerosol dynamics are neglected. This sounds different in the abstract as it is stated that in order "to study effects of aerosol dynamics on the turbulence we vary: : :". Please name explicitly in the methods section not to use aerosol dynamics and state that this is valid because of the short total simulation time used.

On p. 4 210 we mentioned that in this study 'aerosol dynamics" means evaporation/activation of aerosol particles.

Also reformulated:

One should also mention, that in the scope of this model we neglect collision and coalescence of aerosol particles (crucial in creation of rain drops) because of the short total simulation time used.

17) p. 7, l. 380ff: "We find that the number : : : linearly depends: : :." Please check the English and be careful when using three simulations only. Especially Table 3 (p. 5) contradicts. Better skip that sentence or perform more simulations in more narrow Ntot steps.

Reformulated

We find that that at t= 3 s the number of activated particles proportional to the total number, whereas the change of Ntot by a factor of hundred increases LWC by approximately 40 % (Table3}).

18) p. 7, l. 400f: "We find that the vertical motion of air is decelerated because of aerosol dynamics." This contradicts to the statement of neglecting aerosol dynamics (condensation and coagulation) during the period of simulation (p. 7, l. 353)! Please check.

We assume that aerosol dynamics includes condensation and evaporation (it was mentioned in the text). Coalescence and collisions are neglected. Indeed, vertical motion is accelerated because of condensation: the temperature inside the domain decreases, the difference between temperatures inside and outside the domain increases and the buoyancy force also increases.

19) p. 8, l. 405ff: You explain the air temperature change driven by the condensation of water vapor onto the aerosol particles and the release of latent heat. But since the aerosol particles are rather huge size shouldn't matter and the condensation should occur independent on the number if any particle number and time are available. But the change differs notably between 55 and 550 cmôĂĂĂ3 and I can only think of not sufficient time for condensation.

We added a new figure to show time dependence of supersaturation averaged over domain. In this Fig. we present the supersaturation averaged over domain for {\it cases 1, 2, 3 and 4} (see Table 1) and analyze the phase relaxation time of supersaturation t_phase for different values of initial supersaturation and total number of particles. Analyzing numerical results we get the relaxation time of about 0.77 s (case 2), 0.17 s (case 3), and 0.6 s ( case 4). In case 1 the phase relaxation time is larger than time of simulations. We fit the corresponding curve by exp(-t/tau) and get t_phase=4 s.

20) p. 8, l. 436f. The information on the model sizes is very nice but would be best to include it earlier in the methods section for a better understanding on set-up and interpretation of results.

done

21) p. 8, l. 450ff: Very nice indeed. But simulating a 10x10x10 cm3 volume this would cause dramatic horizontal and vertical gradients and motion. Is this still applicable by the present method including the problematic areas along the edges of the finite volume?

To analyze the effect of aerosol and droplets on turbulence a small volume with supersaturation of 10% was considered. Under such extreme conditions, condensation was the dominant process. The results cannot be linearly extended to bigger cloud volumes but should be considered as relevant for a small cloud parcel with extreme supersaturation due to turbulent mixing of the water vapor and temperature field.

22) p. 8, end: Very nice and interesting results indeed. I would recommend a short statement to potential implications for cloud simulations and weather prognosis. This would definitely increase the range of potential readers, for which the area is highly relevant.

At the end of the manuscript we added

We conclude that aerosols quite strongly influences the dynamics in the cloud area. Such effect of aerosols can be crucial also for large scales usually studied with Large Eddy Simulation (LES) and the LES parametrization can be improved with direct numerical simulations.

Indeed, large eddy simulations are based on parametrization of dynamical coefficients, for example, viscosity and diffusion coefficient. In turn, we show in the manuscript that aerosol makes strong effect on air motion. Therefore, aerosol particles most probably strongly affect the dynamical coefficients and more detailed study of this problem is needed.

---

## Author Comment (AC2) · 20 May 2016

This article studies the effect of aerosol dynamics on atmospheric small-scale turbulence using direct numerical simulations. As I already pointed out in my original assessment of the article, my main concern with this article are the extreme initial conditions that were chosen for the simulations: While I understand the concept of fluctuations and the concurrent possibility of achieving extreme values, it is very hard for me to assess how relevant it is to study such an extreme case outside of that context. To elaborate on what I mean, let's take the article by Kulmala et al. that has also been cited by the authors: Kulmala et al. Treat the saturation ratio (let's call it S0 here, because S is used for the supersaturation in the present article) as a stochastical variable with a Gaussian distribution around an average value which varies from 0.995 to 1.0 with a standard deviation of up to 0.05. They then conduct a series of simulations

where they allow the saturation ratio to vary randomly according to the assumed distribution and find that particles can activate also in under-saturated conditions due to the temporal fluctuations in the saturation ratio. In the present paper, the authors pick one very extreme case out of this distribution, which corresponds to a saturation ratio of 1.1 or a supersaturation of 10 % (I can only guess that they still assume the average $S0$ in the cloud to be equal to one). Just to put this into context, common supersaturation values at the base of a cloud are of the order of 0.1 to 0.5 %; according to a quick test conducted with a cloud parcel model that does not consider fluctuations, it reqires a particle number concentration of 1 cmôÃĂĂ3 and an updraft velocity of 10 m/s to achieve a supersaturation of 10 %. Furthermore the authors chose a very high temperature difference between the simulation domain and its surroundings, which is not motivated in the text at all. According to these extreme initial conditions, the authors then also find that aerosols have a strong influence on turbulence, but I wonder how justifiable such a conclusion is without also considering more moderate supersaturations which are, after all, much more likely to occur. Furthermore, I am a little bit skeptic how reliable the results presented here are, as the simulations include the use of random numbers (this especially concerns the generation of the initial turbulence) and thus a single simulation may not be very representative of an average behaviour.

To conclude, I cannot recommend this article for publication in the current form. At the very least the paper requires one more set of reference simulations with a more conventional supersaturation of, say, 0.3 %, and a proper discussion on the issues I layed out above.

According to the referee's comment we have made three additional runs

- the difference between the temperature in the simulation domain and its surroundings of 2.5 K and aerosol particles are included

- the difference between the temperature in the simulation domain and its surroundings of 2.5 K and aerosol particles are not included

- humidity is 10 % smaller than it was in the previous simulations and therefore, super-saturation averaged over domain is 0.6

The last section about the effect of aerosol on the turbulent motion is rewritten based on new input parameters.

Concrete remarks

1. The English is not very good and needs to be reviewed. Some of the sentences are very hard to understand.

We double checked the English and rewritten unclear parts.

2. lines 12–14: Latent heat release is a time dependent process, but finally, close to equilibrium, the total cooling depends (nearly) only on the initial amount of super-saturation. It is therefore unnecessary to state that "even small amounts of aerosols increase the air temperature", and it is quite misleading to give the (very high) change in temperature of 1 K without also giving the value for the supersaturation used.

Reformulated:

We find that the even small amount of aerosol particles (55.5 cm-3) strongly affects the air temperature due to release of latent heat.

3. lines 115–120: Why is the chemical composition of the air important to this study? Wouldn't it be enough to state the total water content?

The model used in this study was prepared for problems with more complicated chemistry. We are planning to develop it in future.

4. lines 126–129: Is this the total difference in temperature/water vapor content between the (upper and lower?) edges of the domain (e.g. 0.001 K/10 cm)? According to the units it cannot be a gradient.

Reformulated:

Based on data of CARRIBA observations typical for the upper parts of clouds / cloud edges in a height of 2000 m, we set the initial conditions for air temperature (T0 = 285.4 K) and water vapour mixing ratio (q_0= 0.0124). The small vertical gradients of temperature and water content are also based on the CARRIBA measurements: the total difference between values of air temperature and water vapour mixing ratio at the upper and lower edges of the domain are (Delta T = 0.001 K) and (Delta q= 4 10-5), correspondingly.

5. lines 133–135: How exactly was this range of supersaturation derived?

The take measurements for temperature and absolute humidity (AH). S=(AH*rho*pw/ps-1)*100%

AH=Yw/rho Yw is water vapour mass fraction rho is air density pw=rho RT/18 is water vapour pressure psf is saturation water vapour pressure

rho=9.6 10-4 g/cm3 Yw= 0.0123759 T=285.4 K

6. line 264: I have never heard the word "unequilibrium" before – I believe the correct term is "non-equilibrium".

We agree that the phrase "non-equilibrium" is more common and we change accordingly

7. Figure 1: The velocity fields in panels (a) and (c) look curiously similar, while they are totally different in panel (b). Did you use exactly the same initial velocity fields in all three cases (or probably a different one for Ntot = 555 cm-3)? How much does varying the particle concentration actually affect turbulence?

Velocity field is taken the same in all free cases. Panels (a) and (c) look similar because the positions of warm/cold layer are similar there (comparing with panel (b)). But it was double checked that the difference between (a) and (c) exists.

8. lines 306–317: How does the initial supersaturation vary? Are these the values

that you mention in lines 133–135? If so, it is not obvious how only the temperature is responsible for this variation in supersaturation, as you also set a gradient in water vapour mixing ratio. I think a figure that clearly lays out the initial conditions of the simulations would be very helpful in the overall understandability of this paper.

In the first version of the manuscript such figures were included but the other referee asked to remove them because in his opinion they were not informative.

9. lines 323–326: Can you somehow quantify how long it would take the system to equilibrate, "more than 3 s" is pretty vague?

We added a new figure to show time dependence of supersaturation averaged over domain for Ntot=55.5, 555 and 5550.

In this Fig. we present the supersaturation averaged over domain for {\it cases 1, 2, 3 and 4} (see Table 1) and analyze the relaxation time of supersaturation tr for different values of initial supersaturation and total number of particles. Analyzing numerical results we get the relaxation time of about 0.77 s (case 2), 0.17 s (case 3), and 0.6 s ( case 4). In case 1 the relaxation time is larger than time of simulations. We fit the corresponding curve by the function exp(-t/tau) and estimate the relaxation time of 4 s.

10. lines 341–347: As the condensational growth of particles is limited by the total surface area, it is not very surprising that the water vapor mixing ratio decreases faster if more particles are present. Also, as for this reason the supersaturation is always larger for smaller number concentrations, it should be expected that these particles grow larger. I don't know if this really "confirms the Twomey effect" or if it rather confirms that the model works as it should.

Removed the sentence about Twomey effect

11. line 359: Replace "number" with "fraction".

replaced

12. Figure 5: Why does the fraction of activated particles go down after about half a second for Ntot = 55:5 cm-3?

This is most probably the effect of nonlinearity of activation process and periodic BC. There are no such fluctuations in other two cases because in these two cases the system equilibrates very fast.

13. lines 380–384: I don't understand the relevance of this statement. As you have the same total water content in all three cases to begin with, the difference in LWC very strongly depends on the time at which you compare the simulations (apart from some small corrections due to temperature and droplet size), because in some simulations the droplets still grow while in others they don't. How would this difference change if you waited 6 instead of 3 seconds?

Added to the text that this result depends on time

We find that at t= 3 s the number of activated particles is proportional to the total number, whereas the change of Ntot by a factor of hundred increases LWC by approximately 40 %.

14. lines 385–399 and Figure 6: How exactly does the buoyancy force affect the turbulence in the simulation domain at scales smaller than 10 cm? This is not discussed at all in the manuscript. How valid is the assumption of a temperature difference of 7.6 K between the simulation domain and its surroundings, especially when the cloud parcel is colder than the rest? From Figure 6 I roughly estimate a deceleration of the air parcel of 25 cm s2 (solid black line), so the air parcel drops more than 1 m (more than ten times the domain size) during the 3 s simulation, still the environment temperature never changes. How does this conform with the original statement that the temperature in the cloud is subject to strong fluctuations? On a side note, if the temperature changes by 1 K, it is not surprising that the buoyancy force changes quite a bit and I wonder if Figure 6 is necessary at all – why not just give a value for B in the different cases? Furthermore, what does the comparison of the two simulations tell us.

No aerosols means no cloud – should you rather compare between different aerosol loadings to somehow mimic what could happen at the edge of the cloud?

As it was mention at the beginning of this respond we have prepared results of simulations in a case of smaller difference between the temperature inside the domain and its surroundings (delta T= 2.5 K). Moreover, now the temperature inside the domain is larger than outside the domain and we get updraft instead of downdraft. We agree that new parameters are more realistic in atmosphere. However, our general conclusion concerning the effect of aerosol on turbulence does not change. We showed in the last section that the aerosol affects the turbulence through the buoyancy. However, since now temperature difference changes sign the vertical air motion is accelerated (rather than decelerated as it was before) if aerosol particles are present.

15. lines 400–410: To my eye the TKE in Figure 7 only differs significantly between aerosols and no aerosol during a short period of time around 2 s. Accordingly, Figure 8 shows a strong jump from roughly zero up to 80 % at that point in time and starts dropping off again thereafter. Still, from Figure 8 the authors conclude that aerosols affect turbulence strongly. How sure are you that these results are not sensitive to the specific set of random numbers used for the simulation (i.e. what would happen if you used a different random seed in the setup)?

We agree that to make the general conclusions about the effect of the turbulence one set of parameters are not sufficient, and since the turbulence is very complicated process some statistics is needed. But the DNS are very computationally demanded and even one set of parameters needs huge amount of time and resources. In this study we illustrate how aerosol important or not important for correct description of turbulent motion, taking conditions typical for atmosphere.

16. lines 406–410: This sentence is very hard to understand. Do you mean to say that the temperature difference between the domain and its surroundings decreases because the temperature inside the domain increases?

Yes.

But now we take warmer domain and cooler its surroundings and instead of decrease/deceleration we get increase/acceleration effect.

This part is reformulated:

The air temperature increases because of release of latent heat caused by condensation onto drops, and therefore, the difference between temperatures inside and outside the domain is enlarged. It results in increased buoyant force and acceleration in vertical direction.